# IL-3-Induced Immediate Expression of c-*fos* and c-*jun* Is Modulated by the IKK2-JNK Axis

**DOI:** 10.3390/cells11091451

**Published:** 2022-04-25

**Authors:** Hirotaka Fujita, Toshitsugu Fujita, Hodaka Fujii

**Affiliations:** Department of Biochemistry and Genome Biology, Hirosaki University Graduate School of Medicine, 5 Zaifu-cho, Hirosaki 036-8562, Aomori, Japan; h19gm101@hirosaki-u.ac.jp (H.F.); hodaka@hirosaki-u.ac.jp (H.F.)

**Keywords:** IL-3, IKK, immediate early gene, JNK

## Abstract

Interleukin (IL)-3 is a pleiotropic cytokine that regulates the survival, proliferation, and differentiation of hematopoietic cells. The binding of IL-3 to its receptor activates intracellular signaling, inducing transcription of immediate early genes (IEGs) such as c-*fos*, c-*jun*, and c-*myc*; however, transcriptional regulation under IL-3 signaling is not fully understood. This study assessed the role of the inhibitor of nuclear factor-κB kinases (IKKs) in inducing IL-3-mediated expression of IEGs. We show that IKK1 and IKK2 are required for the IL-3-induced immediate expression of c-*fos* and c-*jun* in murine hematopoietic Ba/F3 cells. Although IKK2 is well-known for its pivotal role as a regulator of the canonical nuclear factor-κB (NF-κB) pathway, activation of IKKs did not induce the nuclear translocation of the NF-κB transcription factor. We further revealed the important role of IKK2 in the activation of c-Jun N-terminal kinase (JNK), which mediates the IL-3-induced expression of c-*fos* and c-*jun*. These findings indicate that the IKK2-JNK axis modulates the IL-3-induced expression of IEGs in a canonical NF-κB-independent manner.

## 1. Introduction

Interleukin (IL)-3, mainly produced by activated T cells and mast cells, is a key cytokine that regulates the survival, proliferation, and differentiation of hematopoietic cells [1,2]. Although IL-3 is not essential for steady-state hematopoiesis, it stimulates the proliferation and differentiation of myeloid cells and modulates the cellular functions of multiple immune cell types in response to infection or inflammation [3,4,5,6,7]. In addition to its role in immune responses, IL-3 is involved in malignant hematopoiesis. The levels of expression of the IL-3 receptor α (IL-3Rα) subunit are elevated in the progenitor and stem cells of patients with several hematological malignancies, e.g., acute myeloid leukemia (AML), chronic myeloid leukemia (CML), and basic plasmacytoid dendritic cell neoplasm [8,9,10,11,12]. Indeed, its levels of expression strongly correlate with a reduced survival rate in patients with AML [12,13], suggesting that targeting IL-3 signaling might be a potential therapeutic strategy for AML. These findings indicate the need to identify and validate mediators of IL-3 signaling.

IL-3 belongs to the β common (βc) cytokine family, including granulocyte-macrophage colony-stimulating factor (GM-CSF) and IL-5. The βc family functions through heterodimeric receptors composed of a cytokine-specific α chain and a shared βc chain. The receptors of the βc cytokine family lack intrinsic kinase activity, but the βc chain associates with Janus kinases (JAKs) at the membrane-proximal cytoplasmic domain [14]. The binding of IL-3 to IL-3Rα triggers the assembly of the receptor complex to activate JAKs, thereby promoting multiple signaling cascades. Major downstream signaling pathways associated with βc receptor cytokine signaling include the JAK-signal transducer and activator of transcription (STAT), the phosphoinositide 3-kinase (PI3K), and the mitogen-associated protein kinase (MAPK) pathways [15,16]. These signaling pathways are associated with various cellular responses, including cell survival, proliferation, and differentiation.

Immediate early genes (IEGs), including c-*fos*, c-*jun*, and c-*myc*, are expressed rapidly downstream of βc signaling in response to IL-3 stimulation [17]. These IEGs encode transcription factors that control the expression of downstream effector genes and play crucial roles in the molecular mechanisms underlying various cellular processes, including the survival and differentiation of myeloid cells and the production of cytokines [18,19,20]. Two distinct signaling pathways involving the cytoplasmic domain of the βc chain have been associated with the induction of c-*fos*/c-*jun* and c-*myc* transcription. In the first pathway, signaling evoked by the membrane-proximal region of the βc chain, essential for phosphorylation of JAK2, induces the expression of c-*myc* and the activation of proliferation signals [21]. Although the expression of c-*myc* induced by βc signaling was thought to be regulated by STAT5, the dominant-negative form or knockdown of STAT5 did not affect the expression of c-*myc* [22,23], indicating a need for further analysis to determine its transcription mechanism. In the second pathway, MAPKs, including extracellular signal-regulated kinase (ERK), c-Jun N-terminal kinase (JNK), and p38, are evoked by the distal region of the βc chain and primarily control the expression of c-*fos* and c-*jun* [21,24,25]. In response to various stimuli, MAPKs have been shown to activate transcription factors, such as serum-response factors (SRFs), ternary complex factors (TCFs)/Elk-1, and activating transcription factors (ATFs), inducing the transcription of c-*fos* and c-*jun* [26,27]. Although these MAPK pathways are evoked via the βc chain, it is unclear whether they apply to IL-3 signaling.

The inhibitor of nuclear factor-κB kinase (IKK) complex, consisting of two catalytic subunits, IKK1 and IKK2, and a regulatory subunit NEMO, has been shown to regulate nuclear factor-κB (NF-κB) signaling. The canonical NF-κB pathway mediated by IKK2 and NEMO induces the degradation of the inhibitor of nuclear factor-κB (IκB) protein and nuclear translocation of NF-κB transcription factors, which interact with specific promoter elements to activate gene expression [28]. In the non-canonical NF-κB pathway, IKK1 mediates nuclear translocation of the RelB/p52 NF-κB complex [29]. In addition to their involvement in the two NF-κB pathways, IKK1 and IKK2 phosphorylate various substrates, thereby being involved in many biological processes, including cell growth, metabolism, apoptosis, and the cell cycle in an NF-κB-independent manner [30,31]. IKKs were associated with IL-3 signaling in regulating the cell survival and proliferation of myeloid cells. In response to IL-3, IKKs prevent cell death by inducing the degradation of p53 upregulated modifier of apoptosis (PUMA) [32]. Furthermore, IKKs promote cell proliferation by activating mitogenic signaling without activating canonical NF-κB signaling [33]. These findings indicate the importance of NF-kB-independent IKKs in IL-3 signaling.

The findings linking IKKs and IL-3 signaling suggest that IKKs can regulate the expression of IEGs to mediate diverse cellular responses. The present study, therefore, investigated the role of IKKs in mediating the transcription of c-*fos* and c-*jun* downstream of IL-3, and showed that this mechanism was independent of canonical NF-κB signaling. These findings may be useful in elucidating the mechanisms underlying IL-3-mediated malignant hematopoiesis.

## 2. Materials and Methods

### 2.1. Cell Culture and Cell Stimulation

Mouse IL-3-dependent hematopoietic cell lines, 32D [34] and Ba/F3 [35], were maintained in a complete culture medium consisting of RPMI 1640 medium (183-02023, Wako, Osaka, Japan) supplemented with 10% fetal bovine serum (FBS), 10 mM HEPES, 1 mM sodium pyruvate, non-essential amino acids, 5 U/mL penicillin, 50 µg/mL streptomycin, 5 µM 2-mercaptoethanol, and 1 ng/mL recombinant mouse IL-3 (PMC0034, Thermo Fisher Scientific, Waltham, MA, USA). PLAT-E cells [36] were cultured in Dulbecco’s modified Eagle medium supplemented with 10% fetal bovine serum (FBS), 5 U/mL penicillin, and 50 µg/mL streptomycin. All cells were cultured at 37 °C in a humidified 5% CO_2_ incubator.

For cell stimulation, cells were washed three times with phosphate-buffered saline (PBS), starved of IL-3 by culturing these cells for 6 h in a complete culture medium without IL-3 (IL-3 depletion medium), and stimulated by adding 10 ng/mL IL-3 or 20 ng/mL tumor necrosis factor (TNF)-α (410MT025/CF, R&D Systems, Minneapolis, MN, USA). Cells were collected by centrifugation and subjected to subsequent experiments. For inhibitor experiments, cells were incubated for 6 h in an IL-3 depletion medium containing the indicated concentrations of IKK-16 (13313, Cayman, Ann Arbor, MI, USA), JNK-IN-8 (HY-13319, Med Chem Express, Monmouth Junction, NJ, USA), or DMSO. For IL-3 neutralization experiments, 0.4 µg/mL IL-3 was incubated with 10 µg/mL anti-mouse IL-3 neutralizing antibody (Ab) (503901, BioLegend, San Diego, CA, USA) or control IgG (sc-2026, Santa Cruz Biotechnology, Dallas, TX, USA) at 37 °C in RPMI 1640 for 6 h, and then added to IL-3-starved cells. The final concentration of IL-3 was 10 ng/mL.

### 2.2. Generation of Knockout (KO) Cells by Genome Editing

For clustered regularly interspaced short palindromic repeats (CRISPR)-mediated genome editing of IKK1 or IKK2, single guide RNA (sgRNA) sequences for *Streptococcus pyogenes* CRISPR/Cas9 were designed using CRISPRdirect [37]. To construct sgRNA plasmids, complementary oligonucleotides for gRNAs (Appendix A) were annealed and cloned into the *Bbs* I-digested gRNA cloning vector *Bbs* I ver. 2 (85586, Addgene, Watertown, MA, USA) [38]. 

To generate IKK1 or IKK2 single KO clones, 5 × 10^6^ parental Ba/F3 cells were electroporated with 120 µg of a Cas9 expression plasmid (41815, Addgene), 120 µg of a sgRNA expression plasmid targeting *IKK1* or *IKK2*, and 1 µg of a plasmid encoding a puromycin resistant gene using the Gene Pulser Xcell (1652660J1, Bio-Rad, Richmond, CA, USA). For IKK1 and IKK2 double KO (DKO) clones, IKK1 KO Ba/F3 clones were electroporated with the *IKK2* targeting sgRNA expression plasmid together with the Cas9 expression plasmid and the plasmid encoding the puromycin resistant gene. One day after electroporation, 1.0 µg/mL puromycin was added, and cells were cultured for 2 days. The cells were subsequently cultured in a puromycin-free complete medium, and each cell was isolated by limiting dilution. After the expansion of each cell, the knock out of each target gene was confirmed by genotyping PCR followed by Sanger sequencing and immunoblot analysis.

### 2.3. RNA Extraction, Reverse Transcription (RT), and Quantitative PCR

Total RNA was extracted with Isogen II (317-07363, Nippon Gene, Tokyo, Japan) and reverse transcribed using ReverTra Ace qPCR RT Master Mix with a genomic DNA remover (FSQ-301, Toyobo, Osaka, Japan) in accordance with the manufacturer’s protocols. The complementary DNA was subjected to quantitative real-time PCR using THUNDERBIRD SYBR qPCR Mix (QPS-101, Toyobo) and the primers shown in Appendix A. The level of expression of each mRNA was normalized relative to that of *18S rRNA*.

### 2.4. Plasmid Construction and Retroviral Transduction

The plasmid encoding FLAG-tagged IKK1 was purchased from Addgene (15467) [39]. To construct the retroviral plasmid encoding FLAG-IKK1, the coding sequence of FLAG-IKK1 was amplified by PCR using mIKK1-InFusion-F and mIKK1-InFusion-R primers (Appendix A). The PCR product was purified and subjected to In-Fusion cloning (639648, Takara Bio, Shiga, Japan) with the pMSCVpuro vector (634401, Takara Bio) that had been digested with *Bgl* II and *EcoR* I. The sequence of the cloned fragment was confirmed by Sanger sequencing. To construct the retroviral plasmid encoding 3 × FLAG-BAP, the coding sequence of 3 × FLAG-BAP was cleaved by *SnaB* I and *Sma* I from the p3 × FLAG-CMV-7-BAP plasmid (C7472, Sigma-Aldrich, St. Louis, MO, USA) and ligated into the pMSCVpuro vector that had been cleaved by *Hpa* I.

For retrovirus production, the packaging cell line PLAT-E was transfected with 10 µg of the retroviral plasmids in a 10 cm culture dish using Lipofectamine 3000 (L3000015, Thermo Fisher Scientific). After incubation for 48 h, the viral supernatant (10 mL) was collected and filtered through a 0.45 µm pore size membrane. IKK1-deficient Ba/F3 cells were cultured with the viral media supplemented with 10 µg/mL polybrene and 1 ng/mL IL-3 for 5 h, and then an equal volume of complete medium was added. Following incubation overnight, the viral media were replaced with a complete medium containing 1.0 µg/mL puromycin to select and expand infected cells.

### 2.5. Cell Lysis, Immunoprecipitation, and Immunoblot Analyses

Cells were washed with PBS and incubated on ice for 30 min in lysis buffer (20 mM Tris-HCl, pH 8.0, 420 mM NaCl, 1.5 mM MgCl_2_, 1 mM EGTA, 1% Triton X-100, 10% glycerol, 10 mM NaF, 1 mM Na_3_VO_4_, 1 mM Na_4_P_2_O_7_, cOmplete protease inhibitor cocktail without EDTA (11697498001, Roche, Basel, Switzerland). After centrifugation at 20,000× *g* for 10 min at 4 °C, the supernatants were collected as whole-cell extracts. Cytosol and nuclear extracts were prepared with NE-PER Nuclear and Cytoplasmic Extraction reagents (78833, Thermo Fisher Scientific) according to the manufacturer’s protocol with slight modifications, namely, additional incubation with CER II for 2 min at the cytoplasmic protein extraction step and an additional wash of nuclei with CER I before the nuclear protein extraction step. Protein concentrations were determined using the Bradford method with Protein Assay Dye Reagent Concentrate (500-0006, Bio-Rad). 

For immunoprecipitation assays, 5 × 10^6^ cells were lysed in IP lysis buffer (25 mM Tris-HCl, pH 7.5, 150 mM NaCl, 1 mM EDTA, 1% NP-40, 5% glycerol, 10 mM NaF, 1 mM Na_3_VO_4_, 1 mM Na_4_P_2_O_7_, and cOmplete protease inhibitor cocktail without EDTA). The cell lysates were incubated with 3 µg anti-FLAG Ab (F1804, Sigma-Aldrich) or control IgG (sc-2025, Santa Cruz Biotechnology) and 30 µL Dynabeads-Protein G (10003D, Thermo Fisher Scientific) at 4 °C for 1 h. Immunoprecipitants were washed three times with IP lysis buffer and then eluted with 500 µg/mL 3 × FLAG peptide prepared in IP lysis buffer.

The protein samples were boiled for 5 min in a sample buffer (62.5 mM Tris-HCl, pH 6.8, 2.5% SDS, 0.01% bromophenol blue, 5% 2-mercaptoethanol, 10% glycerol), and the proteins were separated by SDS-PAGE and transferred to Immobilon-P membranes (IPVH00010, Millipore, Bedford, MA, USA). The membranes were blocked for 1 h in Tris-buffered saline (TBS) containing 2% bovine serum albumin (BSA) and 0.1% Tween-20 to detect phosphorylated IKK1/2 or PBS containing 5% non-fat milk and 0.1% Tween-20 for the other experiments. The membranes were incubated with primary Ab overnight at 4 °C, washed with TBS containing 0.05% Tween-20, and incubated with the appropriate horseradish peroxidase-conjugated secondary Ab. Proteins were visualized by chemiluminescence using Chemi-Lumi One Ultra (11644, Nacalai Tesque, Kyoto, Japan) to detect phosphorylated IKK1/2, or using ECL prime (RPN2236, GE Healthcare, Pittsburgh, PA, USA) for the other experiments. The Abs used included anti-IKK1 (sc-166231), anti-fibrillarin (sc-25397), anti-IκB-α (sc-1643), anti-JNK (sc-7345), anti-phospho-JNK (sc-6254), HRP-conjugated anti-rat IgG (sc-2065) (purchased from Santa Cruz Biotechnology), anti-phospho-IKK1/2 (2697), anti-p65 (6956) (purchased from Cell Signaling, Danvers, MA, USA), anti-α-tubulin (017-25031), anti-GAPDH (016-25523) (purchased from Wako), HRP-conjugated anti-mouse IgG (NA9310V), HRP-conjugated anti-rabbit IgG (NA9340V) (purchased from GE Healthcare), and anti-IKK2 (MAB7155, R&D Systems). Reproducible results of immunoblot analyses were acquired (at least two times), and a representative result is shown in each figure.

### 2.6. Statistical Analyses

Differences between the two groups were assessed by the Mann-Whitney *U* test. Relative expression levels in multiple groups were compared by Kruskal-Wallis one-way analysis of variance followed by the Steel-Dwass multiple-comparisons test using R software (v3.5.3, R Foundation for Statistical Computing, Vienna, Austria) [40].

## 3. Results

### 3.1. Suppression of IL-3-Mediated Expression of IEGs by an IKK Inhibitor

To determine whether IKK1 and IKK2 are involved in the IL-3-induced expression of IEGs, the effects of IKK-16, a selective inhibitor of IKK1 and IKK2 [41], on IL-3-induced expression of IEGs were tested. IL-3-dependent cell lines, Ba/F3, and 32D were utilized in this study. These cell lines have been extensively studied as models to investigate IL-3 signaling because of their steady expression of the IL-3 receptor and signaling molecules. Ba/F3 or 32D cells were starved of IL-3 for 6 h in the presence of IKK-16 or DMSO (negative control). Upon IL-3 stimulation, expression of c-*fos*, c-*jun*, and c-*myc* was induced rapidly in the absence of IKK-16 in Ba/F3 cells (Figure 1A). By contrast, IKK-16 (0.25 µM) strongly suppressed the IL-3-induced expression of c-*fos* and c-*jun* but not c-*myc* (Figure 1A). Similarly, IL-3-induced expression of c-*fos* was suppressed by 0.25 µM IKK-16 in 32D cells; however, that of c-*myc* was not (Figure 1B). In 32D cells, c-*jun* expression was not detectable [42]. Lower concentrations of IKK-16 (≤0.25 µM) also suppressed c-*fos* and c-*jun* (in Ba/F3 cells), but not c-*myc*, expression 30 min after IL-3 stimulation. By contrast, higher concentrations of IKK-16 (≥0.5 µM) decreased the expression of all these genes (Figure 1C and Appendix A). These results suggest that IKK-16 suppresses two distinct signaling pathways at different concentrations, one that induces c-*fos*/c-*jun* and another that induces c-*myc*. Cell-free assays have shown that IKK-16 inhibits kinase activities of IKK1 and IKK2 at different concentrations (IC_50_: 0.2 and 0.04 µM, respectively) [41]. Therefore, the different sensitivities of the c-*fos*/c-*jun* and c-*myc* pathways to suppression by IKK-16 may be due to the differential contributions of IKK1 and IKK2 to the mechanisms of induction of these IEGs, or to an inherent off-target effect of IKK-16. We hereafter used Ba/F3 cells expressing c-*fos*, c-*jun*, and c-*myc* to investigate the roles of IKKs in the IL-3-mediated expression of different IEGs.

### 3.2. KO of IKKs Suppressed IL-3-Mediated Expression of IEGs

To identify the IKK isoforms responsible for IL-3-mediated expression of IEGs, IKK1 KO, IKK2 KO, and IKK1/2 DKO cells were generated by genome editing using the CRISPR/Cas9 system. The expression plasmids for sgRNAs that target nucleotide sequences encoding the N-terminus of the IKK1 or IKK2 protein were co-transfected, along with a Cas9 expression plasmid to induce frameshift mutations. The absence of IKK1 and/or IKK2 proteins was confirmed by immunoblot analysis (Figure 2A).

Assessments of the levels of expression of c-*fos*, c-*jun*, and c-*myc* mRNAs after IL-3 stimulation showed that KO of IKK1 and/or IKK2 reproducibly led to a reduction in the level of c-*fos* mRNA when compared with parental cells, suggesting that IKK1 and IKK2 are functionally important for IL-3-induced expression of the c-*fos* gene (Figure 2B and Appendix A). In addition, KO of IKK1 alone led to a reduction in expression of c-*jun* mRNA, whereas only one IKK2 KO clone showed a reduction in expression of c-*jun* mRNA compared with parental cells (Figure 2C). Further analysis using two additional IKK2 KO clones showed similar results (Appendix A). Although we cannot exclude the possibility that the variation in c-*jun* mRNA expression in IKK2 KO clones is due to clonal variation, compensatory mechanisms, such as functional redundancy, could mask the loss-of-function phenotypes of some clones. Together with evidence showing that c-*jun* expression was suppressed by a low concentration of IKK-16 (Figure 1A,C), these findings suggest that IKK2 could be involved in the induction of c-*jun* transcription. In contrast to c-*fos* and c-*jun* mRNAs, the expression of c-*myc* mRNA was not affected by either IKK1 or IKK2 KO (Figure 2D). A slight reduction in c-*myc* mRNA expression was observed in IKK1/2 DKO cells, suggesting a functional redundancy between IKK1 and IKK2 in the mechanism of c-*myc* expression (Figure 2D).

In summary, these findings showed that (1) IKK1 is required for IL-3-induced immediate expression of c-*fos* and c-*jun*, (2) IKK2 is also required for expression of c-*fos* and c-*jun*, although a marked difference in clones was observed, and (3) both IKK1 and IKK2 could be marginally involved in the expression of c-*myc*. The finding that Ab neutralization of IL-3 suppressed expression of IEGs (Appendix A) suggests that their expression was induced by IL-3 associated signaling, not by potentially contaminating lipopolysaccharide (LPS) or other Toll-like receptor ligands. Furthermore, these results of loss-of-function experiments suggest that, at concentrations of higher than 0.5 µM, IKK-16 has inherent off-target effects on in vivo expression of IEGs.

### 3.3. Regulation of IL-3-Mediated Expression of IEGs by Activated IKKs in a Canonical NF-κB-Independent Manner

IKK2 is the main regulator of the canonical NF-κB pathway and phosphorylated (activated) following the exposure of cells to inducing agents, such as TNF-α and LPS. This, in turn, induces the degradation of IκB and translocation of NF-κB transcription factors from the cytoplasm to the nucleus, leading to the activation of the expression of NF-κB target genes [28,43]. In addition, IKK-dependent but canonical NF-κB-independent induction of gene expression has been reported [44,45]. This study, therefore, evaluated whether IL-3 induction of IKK-dependent expression of IEGs involves canonical NF-κB signaling. To confirm that IKK1 and IKK2 were activated by IL-3, lysates were prepared from IL-3-stimulated Ba/F3 cells, and phosphorylation of IKK1 and IKK2 was evaluated by immunoblot analysis. Consistent with previous findings [32,33], phosphorylation of IKKs was induced within 5 min after IL-3 stimulation (Figure 3A). Although several additional bands were detected, the specificity of the phospho-IKK1 and phospho-IKK2 bands was confirmed using the lysates of IKK KO cells (Figure 3B). Signals of phosphorylated IKK1 were weak; therefore, phosphorylation of IKK1 in response to IL-3 was further confirmed by immunoprecipitation followed by immunoblot analysis. To this end, FLAG-tagged IKK1 (FLAG-IKK1) was expressed in IKK1 KO Ba/F3 cells. After IL-3 stimulation, FLAG-IKK1 was immunoprecipitated with anti-FLAG Ab. Immunoprecipitated FLAG-IKK1 was phosphorylated 5 min after the stimulation (Appendix A). In addition to FLAG-IKK1, endogenous IKK2 that co-immunoprecipitated with FLAG-IKK1 was also phosphorylated after the stimulation (Appendix A), which is consistent with the results presented in Figure 3A,B. Ab neutralization of IL-3 reduced the phosphorylation of IKK1 and IKK2 (Appendix A), indicating that the signaling cascade evoked by IL-3 is responsible for IKK1 and IKK2 activation. Notably, FLAG-IKK1 restored the expression levels of c-*fos* and c-*jun* in IKK1 KO cells (Appendix A), confirming that IKK1 is involved in the expression of IEGs.

We next investigated if the canonical NF-κB pathway was activated during IL-3 signaling. IκB-α protein levels in whole-cell extracts were analyzed by immunoblot analysis. TNF-α stimulation, used as a positive control, obviously decreased IκB-α protein levels (especially at 15 min), whereas IL-3 stimulation did not (Figure 3C). We also evaluated the nuclear translocation of the transcription activator p65, a member of the family of canonical NF-κB transcription factors. Following IL-3 stimulation, the cytosolic and nucleic fractions were separated, and p65 was detected by immunoblot analysis. Although TNF-α stimulation increased the amounts of p65 in nuclear extracts, IL-3 stimulation did not (Figure 3D). The constant protein levels of IκB-α and p65 in the nucleus after IL-3 stimulation were consistent with the previous results showing that GM-CSF, which binds to a βc receptor subunit, does not induce phosphorylation and degradation of IκB and nuclear translocation of p65 [46]. These findings suggested that IL-3 signaling immediately activates IKK1 and IKK2 but does not induce the subsequent degradation of IκB and nuclear translocation of p65, a characteristic feature of the canonical NF-κB pathway. Thus, IL-3-mediated expression of IEGs is regulated by activated IKKs in a canonical NF-κB-independent manner.

### 3.4. JNK Mediates c-fos and c-jun Expression Downstream of IKK2

To characterize the IEG expression mechanism controlled by IKKs, we investigated the signaling cascade downstream of IKKs. IKKs mediate the activation of the JNK pathway responsible for IL-3-induced proliferation [33]. The JNK pathway regulates the transcription of c-*fos* and c-*jun* in response to various stimuli [26]. Based on these previous findings, we hypothesized that (1) IKKs activated by IL-3 stimulation phosphorylate and thereby activate JNK, and (2) activated JNK induces expression of c-*fos* and c-*jun*. To test these hypotheses, we evaluated the phosphorylation levels of JNK using parental and IKK KO Ba/F3 cells. In parental Ba/F3 cells, the phosphorylation level of JNK was increased within 10 min after IL-3 stimulation (Figure 4A). In IKK1/2 DKO cells, the phosphorylation level of JNK was attenuated compared with that in parental cells 10 min after IL-3 stimulation, despite these cells having similar JNK expression levels (Figure 4B). Further analysis using IKK1 KO and IKK2 KO cells showed that IKK2 KO, but not IKK1 KO, reduced JNK phosphorylation (Figure 4C,D). These data demonstrate that IKK2 is relevant for JNK activation.

To confirm the regulation of c-*fos* and c-*jun* expression by JNK during IL-3 signaling, Ba/F3 cells were treated with the selective JNK inhibitor JNK-IN-8 [47]. Treatment with a series of different concentrations of JNK-IN-8 decreased c-*fos* and c-*jun* expression levels in a dose-dependent manner, but c-*myc* expression levels remained constant 20 min after IL-3 stimulation (Appendix A). Treatment with a low dose (0.5 µM) of JNK-IN-8 partially or remarkably reduced IL-3-induced c-*fos* and c-*jun* expression (20 min, Figure 4E). By contrast, in the presence of the inhibitor, c-*myc* expression was unchanged or slightly augmented after IL-3 stimulation. These results suggest that JNK mediates the induction of c-*fos* and c-*jun* expression in an IKK2-dependent manner during IL-3 signaling, which is consistent with our hypothesis.

## 4. Discussion

IL-3 triggers multiple signaling cascades that modulate cellular responses, including cell survival, proliferation, and differentiation. However, the mechanisms by which IL-3 regulates multiple signals resulting in distinct outcomes have remained elusive. The Src-family kinase- and calcium-dependent activation of IKKs by IL-3 has been found to prevent apoptosis and mediate mitogen signaling for cell proliferation, suggesting that IKKs are important mediators of IL-3 signaling [32,33]. The present study focused on the role of IKKs in the IL-3-induced expression of IEGs. IKKs immediately activated by IL-3 signaling were found to modulate the expression of IEGs, especially c-*fos* and c-*jun*, in a canonical NF-κB-independent manner. In addition, IKK2 mediated the activation of JNK, resulting in c-*fos* and c-*jun* expression (Figure 5). While the expression mechanism downstream of IKK1 remains elusive, our findings also indicate that IKK1 is involved in IEG expression. The transcription factors encoded by these genes are immediately expressed in response to IL-3 stimulation and modulate transcription cascades. Thus, the regulation of such early response genes by IKKs could be one aspect of the molecular mechanisms underlying late phases of IL-3-induced cellular activities. 

Although IKK2 is the main regulator of canonical NF-κB signaling, activation of IKKs by IL-3 did not induce degradation of IκB and nuclear translocation of p65. The ability of members of the βc cytokine family to activate canonical NF-κB signaling is unclear. Electrophoretic mobility shift assays have shown that GM-CSF or IL-3 stimulation enhanced the DNA-binding activity of NF-κB [46,48,49]. Although IL-3 was found to induce the nuclear translocation of p65 in mouse embryonic fibroblasts [49], the nuclear translocation of p65 and the degradation of IκB were not detected in bone marrow mast cells and immune cell lines [33,46]. Despite the inconsistency among experiment assays and cell types, our findings indicate that IKKs can mediate IL-3 signaling in a canonical NF-κB-independent manner. IKK1 has been found to regulate a non-canonical NF-κB pathway that induces the nuclear translocation of the RelB/p52 NF-κB complex [29]. In contrast to the rapid activation of canonical NF-κB signaling, activation of non-canonical NF-κB signaling is slow due to the requirement for de novo protein synthesis [50,51]. Because the transcription of IEGs occurs within 20 min of stimulation, non-canonical NF-κB signaling was likely unrelated to IL-3-induced IEGs transcription.

A JNK inhibitor abrogated c-*jun* expression and decreased c-*fos* expression (Figure 4E). In general, c-*jun* expression is primarily regulated by JNK through direct phosphorylation and transcriptional activation of c-Jun-ATF2 heterodimers [26]. Therefore, IL-3-induced c-*jun* expression is mediated by the same JNK-dependent pathway. Accumulated evidence indicates that phosphorylation of Elk-1 by ERK plays a pivotal role in the ternary complex formation of Elk-1 with SRF, thereby enhancing c-*fos* transcription [52]. Indeed, IL-3-dependent activation of ERK enhances the transcriptional ability of Elk-1 [53]. Therefore, ERK-Elk-1 signaling may be involved in IL-3-induced c-*fos* expression. On the other hand, JNK can also phosphorylate (activate) Elk-1 in the presence of extracellular stimulation [54,55]. The JNK inhibitor partially suppressed IL-3-induced c-*fos* expression (Figure 4E); therefore, JNK may mediate this expression via phosphorylation of Elk-1 to some extent.

IKK1 phosphorylates various substrates involved in transcription mechanisms. For example, IKK1 phosphorylates histone tails, resulting in chromatin remodeling to recruit transcription factors, followed by the onset of transcription [56,57]. Optimal transcription of c-*fos* and some IEGs in response to epidermal growth factor requires IKK1 to regulate promoter-specific histone H3 phosphorylation independently of IκB degradation in mouse embryonic fibroblasts [44]. These findings suggest that IKK1 activated by IL-3 could lead to the phosphorylation of histone tails, initiating the transcription of c-*fos* and c-*jun*. Other examples of transcriptional regulation by IKK1 are phosphorylation of transcription factors and cofactors, e.g., AP-1, Myc, and CBP, to fine-tune signals for proper gene expression through NF-κB-dependent and -independent mechanisms [30,31,58]. Therefore, known or unknown substrates of IKK1 could regulate the transcription of IEGs in response to IL-3.

IL-3 supports the differentiation of hematopoietic progenitor cells and stimulates various myeloid cells. In addition, IL-3 plays a role in the development and functions of lymphocytes. IL-3 induces the expansion of stimulated B cells [59] and immunoglobulin secretion [60]. Activated T helper cells and regulatory T cells express functional IL-3Rα and exhibit IL-3-induced activation of the JAK-STAT pathway [61,62]. However, little is known about intracellular signaling and the roles of IL-3 in lymphocytes. It will be interesting to study this and to investigate the difference in IL-3 signaling between myeloid cells and lymphocytes. This study utilized IL-3-dependent cell lines, Ba/F3 and 32D; however, there is a general concern about their phenotypic differences from primary cells, similar to other cell lines. Further studies are needed to elucidate the roles of IKK-mediated IL-3 signaling and the underlying mechanisms in more physiological conditions.

The present study provides evidence that IKK1 and IKK2 regulate IL-3-induced expression of IEGs in a canonical NF-κB-independent manner and that JNK is a key mediator downstream of IKK2. In response to IL-3 stimulation, JNK plays a role in cell survival through the inactivation of proapoptotic signaling [63]. Therefore, the IKK2-JNK axis could be responsible for cell survival via IEG expression. These findings may help clarify the molecular mechanisms of cell survival and proliferation induced by IL-3 and those underlying IL-3-mediated malignant hematopoiesis.

## Figures and Tables

**Figure 1 cells-11-01451-f001:**
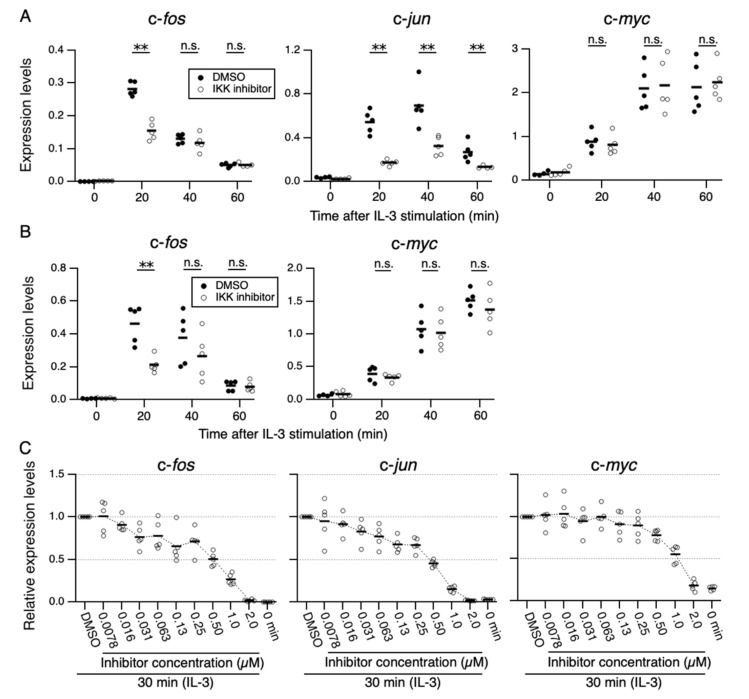
Effects of IKK-16, an IKK inhibitor, on IL-3-induced expression of IEGs. (**A**,**B**) Levels of expression of c-*fos*, c-*jun*, and c-*myc* mRNAs every 20 min after IL-3 stimulation in Ba/F3 and 32D cells (*n* = 5 each). Ba/F3 (**A**) or 32D (**B**) cells were pre-incubated with IKK-16 (0.25 µM) or DMSO for 6 h and then stimulated with IL-3. ** *p* < 0.01; n.s., not significant, as assessed by Mann-Whitney *U*-tests, for differences between IKK-16 treated and control cells at each time point. (**C**) Levels of expression of c-*fos*, c-*jun*, and c-*myc* 0 and 30 min after IL-3 stimulation in Ba/F3 cells in the presence of IKK-16 (7.8 nM–2.0 µM) or DMSO. Each dot represents the result from independent experiments (*n* = 5 each). Means are indicated by black bars.

**Figure 2 cells-11-01451-f002:**
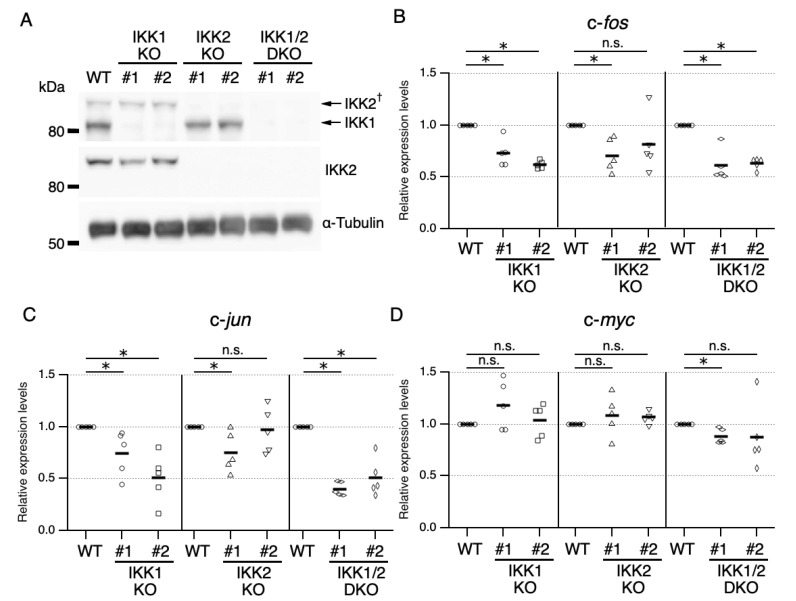
Effects of IKK1 and/or IKK2 KO on IL-3-induced expression of IEGs. (**A**) IKK1 and IKK2 protein levels in Ba/F3 parental cells (WT) and CRISPR/Cas9-mediated IKK1 KO, IKK2 KO, and IKK1/2 DKO cells. Whole-cell lysates were immunoblotted with the indicated Abs. A dagger denotes the detection of IKK2 by potential cross-reactivity of the anti-IKK1 Ab. (**B**–**D**) Relative levels of expression of c-*fos* (**B**), c-*jun* (**C**), and c-*myc* (**D**) mRNAs measured 20, 40, and 40 min, respectively, after IL-3 stimulation of Ba/F3 parental, IKK1 KO, IKK2 KO, and IKK1/2 DKO cells. * *p* < 0.05; n.s., not significant, as assessed by the Kruskal-Wallis test with the Steel-Dwass test. Each dot represents the result from independent experiments (*n* = 5). Means are indicated by bars.

**Figure 3 cells-11-01451-f003:**
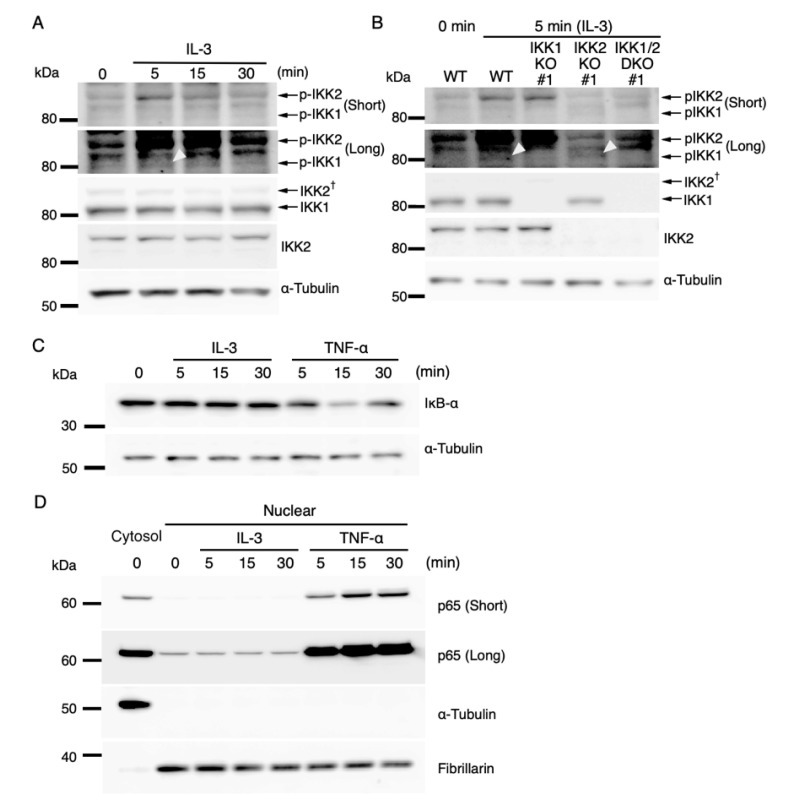
IL-3-induced IKK activation was not associated with the degradation of IκB-α followed by nuclear translocation of p65. (**A**) Phosphorylation of IKKs after IL-3 stimulation. Following IL-3 deprivation for 6 h, Ba/F3 cells were stimulated with IL-3 for the indicated times, and whole-cell lysates were immunoblotted using the indicated Abs. Long and short exposures to detect phosphorylated IKK1 and IKK2 are shown. An arrowhead indicates the band corresponding to phosphorylated IKK1 at 5 min after IL-3 stimulation. A dagger denotes the detection of IKK2 by potential cross-reactivity of the anti-IKK1 Ab. (**B**) Phosphorylation of IKKs in IKK KO cells after IL-3 stimulation. Following IL-3 deprivation for 6 h, Ba/F3 parental (WT), IKK1 KO, IKK2 KO, and IKK1/2 DKO cells were stimulated with IL-3 for 5 min. Immunoblot analysis was performed with whole-cell lysates using the indicated Abs. Arrowheads indicate the bands corresponding to phosphorylated IKK1. (**C**) Degradation of IκB-α by TNF-α but not by IL-3 stimulation. Following IL-3 deprivation, Ba/F3 cells were stimulated with IL-3 or TNF-α for the indicated durations, and whole-cell lysates were immunoblotted using the indicated Abs. (**D**) Induction of nuclear translocation of p65 by TNF-α but not by IL-3 stimulation. Following IL-3 deprivation for 6 h, Ba/F3 cells were stimulated with IL-3 or TNF-α for the indicated times and immunoblotted with Abs to the cytosol marker α-tubulin and the nuclear marker fibrillarin. Long and short exposures to detect p65 are shown.

**Figure 4 cells-11-01451-f004:**
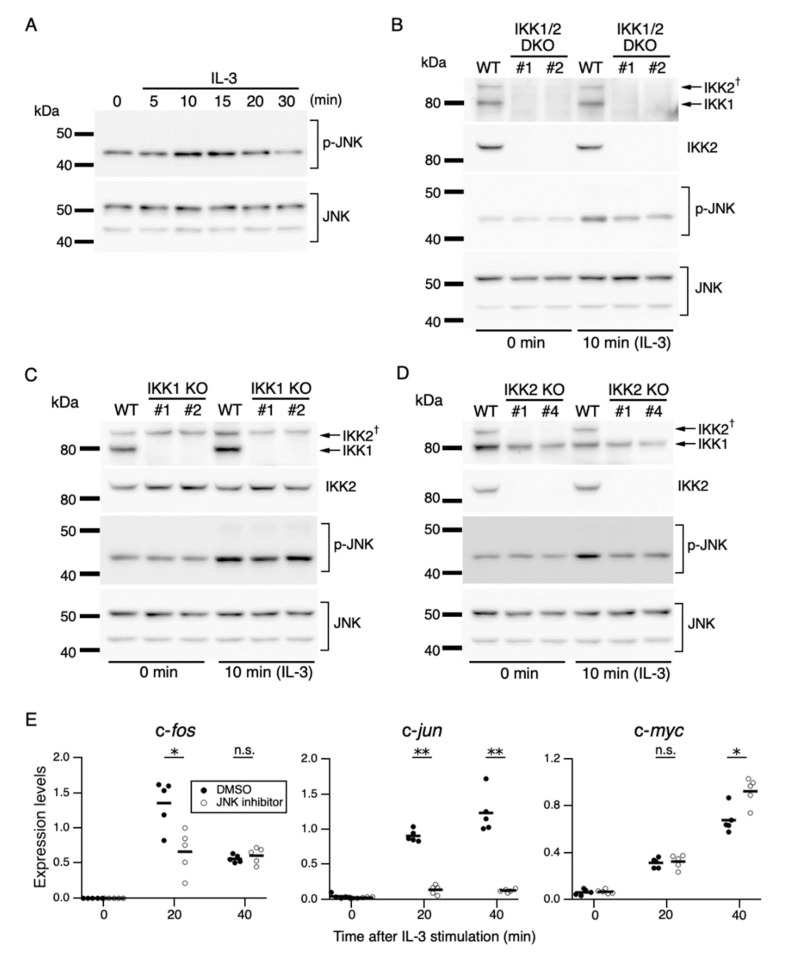
IKK2-mediated activation of JNK regulates c-*fos* and c-*jun* expression. (**A**) Phosphorylation of JNK after IL-3 stimulation. Following IL-3 deprivation for 6 h, Ba/F3 cells were stimulated with IL-3 for the indicated durations, and whole-cell lysates were subjected to immunoblot analysis using the indicated Abs. (**B**–**D**) Phosphorylation of JNK in IKK KO cells after IL-3 stimulation. Ba/F3 parental (WT), IKK1 KO, IKK2 KO, and IKK1/2 DKO cells were stimulated with IL-3 after IL-3 deprivation for 6 h. Immunoblot analysis was performed with whole-cell lysates using the indicated Abs. (**E**) Expression levels of c-*fos*, c-*jun*, and c-*myc* mRNAs 20 and 40 min after IL-3 stimulation in the presence of JNK-IN-8 or DMSO. Ba/F3 cells were pre-incubated with 0.5 µM JNK-IN-8 or DMSO for 6 h and then stimulated with IL-3. * *p* < 0.05; ** *p* < 0.01; n.s., not significant, as assessed by Mann-Whitney *U*-tests, for differences between JNK-IN-8-treated and control cells at each time point. Each dot represents the result from independent experiments (*n* = 5). Means are indicated by bars.

**Figure 5 cells-11-01451-f005:**
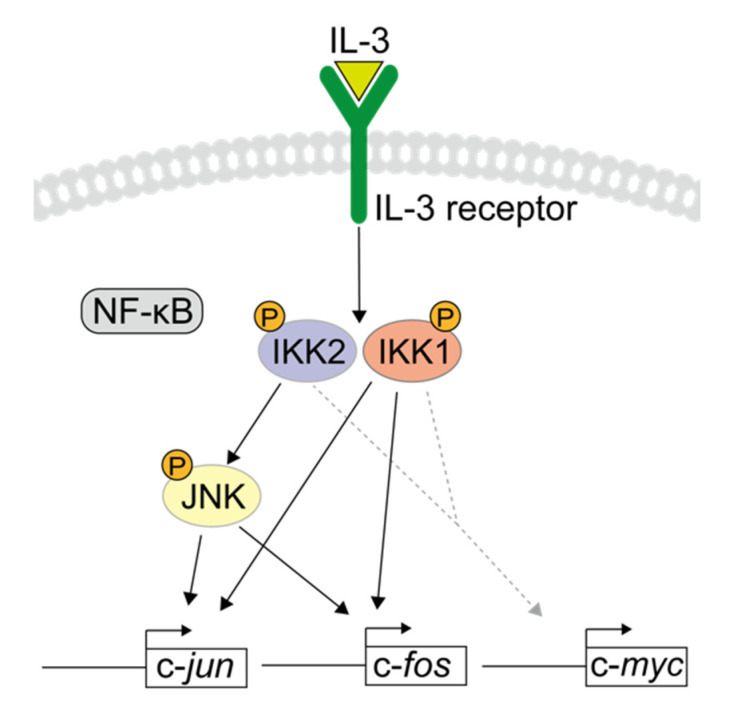
A putative model for IKK-mediated IL-3 signaling to induce expression of IEGs.

## Data Availability

All data relevant to this study are included in the article or Appendix A.

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
