# Peer review of "IL-3-Induced Immediate Expression of c-fos and c-jun Is Modulated by the IKK2-JNK Axis"

_cells, 2022, doi:10.3390/cells11091451_

Round 1

Reviewer 1 Report

I thank oyu for the answers and the addition of these new, nice experiments.

Reviewer 2 Report

I reviewed the research article by Hirotaka Fujita et al., entitled: ”IL-3-induced immediate expression of c-fos and c-jun is modulated by IKK 3” a few months ago. In its present form, I appreciate the changes made by the authors. Specifically, they addressed the role of JNK in this pathway. I would recommend this work for publication in Cells.

This manuscript is a resubmission of an earlier submission. The following is a list of the peer review reports and author responses from that submission.

Round 1

Reviewer 1 Report

In this study, Fujita H. et al., address the role of IKKs in regulating IL-3 mediated expression of the IEG c-fos, c-jun, and c-myc. IL-3, is a member of the βc cytokine and receptor family. Activation of JAK via βc chains leads to c-myc expression, while activation of MAPK regulates c-fos and c-jun. NF-κB is also believed to be involved in IL-3 signalling, so the authors focused on elucidating the involvement of IKKs in IL-3 mediated IEG transcription.

The authors present qPCR and immunoblot data, using Ba/F3 cells, pharmacological IKK inhibitors, and IKK knockout. The study is well written, the presented data are mostly convincing, and the methods and figure legends are described clearly and in detail. Figure S4 is a very appealing addition to this paper, adding to the study’s credibility.

However, in my opinion, there are a few limitations:

1) The authors use only the Ba/F3 cell line. In my opinion, the addition of at least another cell line or primary cells would be essential to verify and validate their key findings.

2) They state that IL-3 activates IKKs, while the NF-κB pathway downstream is not activated, by showing p65 translocation into the nucleus. As this is one of their key findings, further readouts should be added, like IkB degradation or expression of NF-κB target genes to support this hypothesis. Can you provide IF images of p65 translocation (or similar)?

3) The authors show that deletion or inhibition of IKKs lead to reduced expression of c-fos and c-jun, but provide no experimental explanation for this effect. In the discussion they state the possibility of an involvement of IKK mediated JNK or histone H3 phosphorylation, but did not check for this or other possibilities in their study.

4) The usage of the CRISPR/Cas9 system to generate IKK knockout cells is an elegant addition to IKK-16, to describe the consequences of lack or inhibition of IKK during IL-3 signalling. As the key message that IKKs regulate c-fos und c-jun during IL-3 signalling is quite concise, it would be interesting to show a rescue of the knockout phenotype or analyse the effect of an IKK overexpression or the expression of constitutive active versions of IKKs. This may shed light on the fact that the IKK1 knockout cells have reduced c-fos and c-jun (but not c-myc), although there is hardly any IKK1 phosphorylation visible (Figure 3/S3). Activated IKK1 was reported to stabilize c-myc in HEK-293, HeLa and DU145 cells (https://pubmed.ncbi.nlm.nih.gov/33461590/). Thus it may be possible that c-myc expression in your knockout/inhibition system remains stable, because IKK1 phosphorylation may be changed too little, or not at all.

Major comments:

  • Key experiments need to be performed with another cell line, ideally primary cells
  • The authors describe nicely the effects of IKK inhibition or deletion, but provide no experimental explanation
  • Figure 3C: Could you provide further proof of a direct effect of IKKs without the activation of the NF-κB pathway? Could you also provide IF images? Did you detect IkB degradation in your samples?
  • Figure 3A/B and Figure S3: I can see hardly any IKK1 phosphorylation. Especially in the IKK2-KO#1 sample it should be visible. Can you be certain, that IKK1 is phosphorylated/activated during IL-3 signalling?

Minor comments:

  • Ba/F3 cells: describe and discuss advantages and limitations
  • Line 9 and 22: Correct “hemopoietic” to “hematopoietic”
  • Line 17: non-canonical can be confused with the non-canonical NF-κB pathway (= IKK1, RelB)
  • Line 67: You work with both, IKK1 and IKK2. Define which IKK is meant. In canonical it is typically IKK2, in non-canonical IKK1.
  • How do you define a “biological independent sample” using your cell line?
  • Figure 1: Please increase n-number
  • Figure S1B/C: n = 3 but I count 5 dots.
  • Figure S2: n = 5 but I count only 3 dots.
  • Lines 233-236: Again define, which IKK. Canonical NF-κB signalling is via TNFR, IKK2 and IkB. Non-canonical is via (e.g. CD40) IKK1 and p100
  • Figure 3: In addition to these representative blots, increase n-number and show densitometric analysis of IKK1 and IKK2 phosphorylation and nuclear translocation of p65.

Reviewer 2 Report

The research article by Hirotaka Fujita et al., entitled: ”IL-3-induced immediate expression of c-fos and c-jun is modulated by IKK 3,” aims to investigate the role of IKKs in mediating transcription of c-fos and c-jun downstream of IL-3 and showed that this mechanism was independent of canonical NF-κB signaling. The authors suggest that these findings may help elucidate the mechanisms underlying IL-3-mediated malignant hematopoiesis. The article is well written and includes important references. The methods are appropriate and contain details that will allow the reader to reproduce the data. The experiments have all necessary controls and have sufficient explanation. The reviewer appreciates the use of small molecule inhibitors and CRISPR cell lines to demonstrate the results. The data provided by the authors are convincing, specifically, the contrast to TNF-induced nuclear translocation of p65 that is shown side-by-side with IL-3 stimulation. 

Overall, the information is of interest and provides novel evidence about IL-3-induced signal transduction. However, the reviewer would like t make a few suggestions to improve the work and make it more suitable for publication in Cells. 

  1. The findings regarding the regulation of c-jun and c-fos transcription factors in response to IL3 stimulation by IKK in an NFkB-independent manner are of interest for a broad auditory of a signal transduction field. It will be beneficial to attempt to find an intermediate in this novel pathway. C jun is a bona fide downstream target of JNK, and it will be exciting to test if JNK acts downstream of IKK. 
  2. Overall, the article will greatly benefit from experimental efforts to identify an intermediate player downstream of IKK and upstream of c fos and c jun.
  3. IL3 is a marker for leukemic AML stem cells, and the reviewer appreciates the choice of the cell line. However, IL3 also signals in immune cells, such as T cells, and it would be essential to know if the IL3 signaling differs between immune and hematopoietic systems. Please, comment on this or provide an experimental comparison between these two cell types.